# OpenReview forum: "StealthEval: A Probe-Rewrite-Evaluate Workflow for Reliable Benchmarks"
_NeurIPS.cc/2025/Workshop/Reliable_ML — NeurIPS 2025 - Reliable ML Workshop_

### Official Review · Reviewer_Bgsc · 2025-09-19
**Review: StealthEval**

**Rating:** 6
**Confidence:** 3

**Review:**

This paper introduces a workflow to address the following issue in AI alignment, evaluation awareness in large language models. The authors validate that LLMs behave differently in a perceived test environments compared to real-world deployment. This discrepancy can lead to unreliable safety benchmarks. The workflow proceeds as follows,

Probe: A linear probe (a tool for analysing the model's internal representations) is used to score prompts on a continuous scale. A higher score indicates that the prompt appears more like a "deployment" context rather than a "test" context.

Rewrite: Prompts with a low "deployment" score are rewritten by another LLM to be more natural and conversational, while maintaining the original task.

Evaluate: A suite of LLMs are evaluated on both the original and rewritten prompts. An LLM-as-a-judge method is used to classify the responses as honest, deceptive, or a refusal.

The findings indicate a consistent and significant shift in model behavior after the prompts are rewritten. The rewritten, deployment-like prompts led to an increase in honest and refusal responses and a decrease in deceptive responses across most models tested. The work highlights a limitation of current safety benchmarks and suggests that more realistic evaluation methods are necessary to accurately assess true alignment before deployment. However, it is interesting that the shift acts in the opposite direction as one might expect. Existing literature on evaluation awareness suggests that models perform more compliant when being tested. The presented findings contradict this standpoint which I think is crucial to address.

---

### Official Review · Reviewer_28Fa · 2025-09-19

**Rating:** 7
**Confidence:** 3

**Review:**

The paper is built on the following observation: benchmark performance may not accurately reflect a model’s true safety and honesty.  To address this issue, it proposes a systematic method for safety benchmarking evaluation.

The problem is interesting, however, I think the presentation of the paper should be improved. Here are some of my suggestions.

(1) In the introduction part, at least a summary sentence should be added to tell the reader what is the main contribution and focus of this paper.

(2) Section 3.2: Linear probe. First, before using this method, please use simple words to introduce what is "probing". In the current version, we do not know what "Simple Contrastive head at layer 23", " AUROC 0.816 with an operating point near threshold 0.103" are. More importantly, it's important to discuss what role it plays in the total loop, and why adding this part is important for you to solve the problem.

The evaluation and discussion part is very good. However, I think more discussion should be added into Sections 3.4 and 3.5.

Overall, I think it's a good paper.